# Effects of Infill Patterns on the Mechanical and Tribological Behaviour of 3D-Printed Polylactic Acid/Bamboo Biocomposites for Structural Applications

**DOI:** 10.3390/polym17040448

**Published:** 2025-02-08

**Authors:** Devarajan Balaji, Balasubramanian Arulmurugan, Venkateswaran Bhuvaneswari

**Affiliations:** 1Department of Mechanical Engineering, KPR Institute of Engineering and Technology, Coimbatore 641407, Tamil Nadu, India; 2AU-Sophisticated Testing and Instrumentation Centre and Department of Mechanical Engineering, Alliance School of Applied Engineering, Alliance University, Bengaluru 562106, Karnataka, India; bhuvanashankar82@gmail.com

**Keywords:** additive manufacturing, PLA bamboo fiber composite, honeycomb, infill pattern

## Abstract

Composite materials are gaining attention owing to their exemplary characteristics and, if the materials are eco-friendly, they attract much more. One such composite of poly lactic acid (PLA) combined with bamboo fiber in the ratio of 80:20 is selected for this study. The composites are manufactured using additive manufacturing, or the 3D-printing technique. In this article, a novel approach of infilling a honeycomb with around 12 infill patterns has been made, and all the 3D-printed specimens were tested for their mechanical and tribological properties. The 3D-printed composites were characterized using Fourier Transform InfraRed spectroscopy (FTIR) and X-Ray Diffraction (XRD) to evaluate their chemical composition and crystallite size (CS), respectively. Based on the results, the cross infill pattern outperforms irregular geometries like the Gyroid in terms of impact strength owing to its efficient stress distribution and superior interlayer bonding. By utilizing bidirectional reinforcement and distributing loads uniformly, the grid infill was able to attain the Shore D maximum hardness due to its strong 3D lattice structure; the Octet infill is very resistant to wear, which improves energy absorption and decreases material loss. Such honeycomb-filled 3D-printed composites can act as high-mechanical-strength components and find their applications in aerospace applications like drones and their allied structures.

## 1. Introduction

Supporting a part’s shell or skin interior are the main uses of infills in additively manufactured parts. Another way infills are utilized to decrease material volume is by creating a hollow inside the object [1]. Lightweight structures with a greater strength-to-weight ratio can be created using additive manufacturing (AM) by creating prismatic lattice infills without the need for special tooling. The need for interior support for printing makes infills a necessity in many cases when object skins are involved. It is also possible to modify infills to accomplish specific functional goals [2]. There is a close relationship between the object’s physical properties and the infill meso-structure, that is often designed taking into account things like time to build, appearance, deposition consistency, and reinforcement material [3]. The infill meso-structure allows for control over the object’s intrinsic and extrinsic properties. Examples of physical properties examined in relation to the material arrangement patterns include pore size, geometry, and structural integrity [4,5]. A hybrid algorithm for generating contour and zigzag tool pathways was suggested by Jin et al. [6] to decrease build time without compromising geometric accuracy. An approach to multipatch toolpath planning was put forward, which entailed discretizing the design domain into numerous patches and then creating a zigzag tool path inside each patch to guarantee continuous deposition [7]. Achieving the desired infill porosity has been accomplished by the utilization of adaptive layout patterns [8]. A variational deposition pattern for tissue scaffolds based on parametric functions has just been suggested; this pattern attempts to imitate the native tissue’s heterogeneous topology [9]. Topology optimization was suggested by Wu and their fellow researchers [10] as a means of producing porous infill structures that resemble bone.

However, critical manufacturability issues arise in such intricate designs due to things like trapped voids, overhanging features that need support material, and discontinuous tool paths. To create a dense boundary and periodic internal lattice structure, Jiang and their fellow researchers [11] created a topology optimization based on level set for the stereolithography process. A topology optimization approach based on asymptotic homogenization was suggested by Cheng et al. [12] for the design of periodic lattice infill structures with variable densities and improved natural frequencies. In order to generate functional infill patterns, heuristic functions and stress or strain fields are also taken into account. This means that stress values obtained through finite element analysis can be linearly interpolated [13,14]. Discrete infills of varying densities were generated by Roger and Krawczak [15] using the stress field. Mhapsekar et al. [16] improved the production feasibility of the topology-optimized parts. In addition, to manage self-support and surface roughness, Wang and co-researchers [17] limited the slopes of the part boundary during the topology optimization procedure based on desnity. Neither method, however, is suitable for designing part lattice infill structures, since they ignore the components’ internal architecture. Two of the most popular infill patterns are a standard hexagonal honeycomb with a constant density and 0 deg−90° zigzag. When it comes to designing the zigzag infill with continuity, it is not hard at all, and printing it out is as simple as following the vector. A great deal of effort has gone into optimizing the zigzag for various item geometries by manipulating the method parameters [18].

PLA-based biocomposites were tested for their mechanical properties and the influence of infill pattern on the mechanical behaviour was dealt with in much literature. In some of the studies, the authors developed a PLA-Zn biocomposite with zinc at a fraction of 20% and fabricated the biocomposite using traditional 3D-printing techniques. The mechanical behaviour of PLA/Zn biocomposites with different infill patterns were assessed, and the pattern rendering highest mechanical results was evaluated. Using that infill pattern, the authors 3D-printed a drone frame arm and tested it for compressive strength through experimental methods. The infill pattern which mimics the honeycomb structure closely begets better mechanical properties [19]. In some other studies, PLA composites were 3D-printed using various infill patterns and infill percentages using fused filament fabrication (FFF) techniques. The results of non-traditional bending tests portrayed that the honeycomb and Gyroid infill patterns exhibited better behavior owing to their layer orientation. However, anisotropy of the 3D-printed composites was not high for any of the infill patterns, and the crack propagation was retarded by the layer orientation [20]. In various other studies which dealt with infill patterns and percentages for 3D printing of PLA-based composites, it was stated that the printed specimen with a 90° raster angle possessed better mechanical properties. Flat and on-edge orientations rendered better tensile, elongation and fracture behavior, while the vertical orientations showed relatively lesser mechanical properties [21,22].

It could be understood from all the above discussions that the honeycomb structure renders better mechanical properties, while a few other infill patterns which mimic honeycomb also possess better mechanical behavior. But the printing of composites with honeycomb structures leaves substantial airgaps, which may contribute to a reduction in mechanical properties. In order to avoid this, the inherent gaps of the honeycomb structures have to be reduced as far as possible through material filling. Accordingly, in this article a novel approach of infilling honeycomb with around 12 infill patterns is tested for PLA/bamboo biocomposites. The hybrid infill is achieved with the aid of a honeycomb infill further infilled with versatile patterns. The 3D-printed composites were characterized using appropriate methods and tested for mechanical and tribological properties using standard testing procedures.

## 2. Materials and Methods

### 2.1. Materials

PLA infill with natural fiber (bamboo) in the ratio of 80:20 was brought from M/S. AMOLEN, Los Angeles, CA, USA as shown in Figure 1. Table 1 depicts the general properties of the PLA 3D printing grade polymer.

### 2.2. 3D Printing of Composites

The 3D printer Creality Ender 3 V3 SE printer (as shown in Figure 2) was used to print the PB composite filament. The process parameters used for the 3D printing of PB composites are depicted in Figure 2. The infill patterns are Quadra Cubic, Grid, Tri-hexagon, Cubic Subdivision, Concentric, Octet, Cross, Triangle, Zig Zag, Cross 3D, Gyroid, and Cubic. All the specimens were 3D-printed under ambient conditions with the Brim option enabled to avoid warpage of the specimen. This operation enhances the adhesion between the PB filament and the 3D printer bed, facilitating uniform cooling of the layers before the next layer is deposited. The same was also reported in some of the earlier research [19].

### 2.3. Expeirmentation

#### 2.3.1. Mechanical and Tribological Testing

PLA/bamboo filaments were used for the 3D printing of composites with different infill patterns, which were then subjected to mechanical and tribological properties testing. Table 2 shows the ASTM standards adopted for preparing the test specimens and their respective specimen size. The conditions used during testing of the specimens were also enunciated in the table. All the tests were carried out under ambient conditions. A total of five trials were carried out for mechanical testing, while three trials were carried out for tribological testing. Specific wear rate (SWR) was calculated (in mm^3^/Nm) based on the load (P), speed (N), sliding distance (S) and wear volume loss values (ΔV) using the following equation:(1)$$SWR=\frac{∆V}{P\times S}$$

Tribological tests were carried out in a CONMAT pin-on-disc tester. The value of frictional force and the height loss of the composite samples were taken from tribometer interface as such, and the SWR values were calculated based on the obtained values from the machine.

#### 2.3.2. Characterization

The purchased PLA/bamboo filaments were characterized using X-ray diffraction and Fourier Transform Infrared Spectroscopy (FTIR). XRD analysis was used to evaluate the crystallite size of the composites in a Bruker D8 advance machine in between Bragg’s angle (2θ) 5° and 80° using Cu-Kα radiation with a 0.154 nm wavelength at a standard scan rate of 2°/min. The peak data were analyzed, and the crystallite size (CS) was deduced from the peak data using the following Equation (2).(2)$$\mathrm{CS}=\frac{K\lambda }{\beta \cos\theta }$$

FTIR was used to evaluate the chemical constituents of the 3D-printed PLA/Bamboo composite samples in a Bruker model spectrophotometer within a wavelength range of 4000–400 cm^−1^ in attenuated total absorption mode and 4 cm^−1^ resolution. A scanning electron microscope (SEM) (JEOL JSM-7700F FE-SEM model) was used to observe the morphology of the fractured specimen.

## 3. Result and Discussion

### 3.1. Characterization of PLA/Zn Filaments

#### 3.1.1. Fourier Transform Infrared Spectroscopic (FTIR) Analysis

FTIR spectra of the PLA/bamboo composites, as shown in Figure 3, shows that the contain strong absorption bands that correspond to the functional groups of PLA and bamboo. Peaks in the 4000–3000 cm^−1^ range indicate O-H and N-H stretching. In bamboo, cellulose and lignin contribute to the hydroxyl (O-H) vibrations around 3400 cm^−1^. While the range of 3000–2800 cm^−1^ represents C-H stretching from aliphatic chains, a strong peak at 1750–1735 cm^−1^ indicates the ester carbonyl (C=O) stretch in PLA. The polymer matrix and bamboo fibers’ unique vibrational modes are captured in the area below 1000 cm^−1^, whereas the bending and stretching of C-H bonds and C-O bonds respectively are indicated by the peaks between 1500 and 1000 cm^−1^, respectively. There are significant functional groups that are indicative of both materials, as shown in Figure 3.

Table 3 shows the prominent functional groups deduced from the FTIR peaks for PLA/bamboo biocomposites.

#### 3.1.2. X-Ray Diffraction (XRD) Analysis

The crystallinity of the PLA/bamboo biocomposite filament (23.08 nm) is higher, which results in more distinct and sharp peaks, as shown in Table 4. The regular and organized atomic structure of crystalline materials is what causes them to exhibit this sharp peak. A smaller value of CS of the 3D-printed composites conveys that the composites contained condensed particles, resulting in densely packed particles within the filament. Hence, the composites might result in enhanced mechanical properties. Greater scattering power and, perhaps, improved sample alignment or ordering are indicated by the uniformly increased intensity in the PLA/bamboo composite filament specimen. It can be inferred that the two materials share similar structural patterns or crystalline phases because they both exhibit similar peak positions around specific 2θ values, for example, 20° to 30°. It may be due to variations in processing or composition if the number of peaks and peak intensities vary across samples. A similar behavior was observed by some of the previous researchers [26,27].

### 3.2. Mechanical Behaviour of PLA/Bamboo Composites

#### 3.2.1. Impact Properties

An impact test was carried out in an Izod impact tester, with the ASTM impact specimen standard being followed for additive-manufactured specimens. The following Figure 4 shows the specimen of cross infill 3D-printed specimen after the test. In addition, Table 4 reveals the values of impact energy absorbed in J/cm^2^. It reveals that the impact strength of cross infill honeycomb gives higher strength. It is inferred that the cross infill pattern’s intersecting lines form a crisscross pattern that distributes stress along two main directions. This pattern occurs at regular intervals; because this shape is better able to withstand loads pulled in different directions, there is less likelihood of localized failure. Due to its lack of abrupt directional changes, the cross pattern minimizes the number of stress concentration points compared to patterns with more complex or irregular geometries, such as the Gyroid or Tri-hexagon. This distributes the stress evenly within the infill structure. Layers consistently overlap in superior interlayer coupling, which is characterized by a cross pattern that typically improves interlayer adhesion. This is fantastic for composites like PLA + bamboo that rely on strong layer-to-fiber bonding.

Figure 5 shows the morphological analysis of the 3D-printed PLA/bamboo composites after impact failure. As the result depicts, the specimen with the cross-type infill pattern has been taken for analysis of morphology. The image clearly depicts the wall of the honeycomb structure, and the material filled inside the honeycomb structure. The morphology of the specimen shows the uniform thickness of the layers produced during the 3D printing in horizontal and vertical directions. The deposition of the cross-infill pattern within the hexagonal box of the honeycomb structure could also be witnessed in morphology. It could be seen in morphology that a significant amount of air voids is present in the walls of the honeycomb structure, which may affect the mechanical properties. In the current experimental work, due to the filling of the honeycomb, mechanical behavior was enhanced. Moreover, the failure of the specimen was largely due to the layer pullout, which depicts the uniform cooling achieved between the layers during the 3D printing process. Due to the better compatibility between the PLA and bamboo particles, the failure initiation from crater or interface could not be observed in morphology. In addition, layer-to-layer debonding and layer delamination were not observed, depicting the better interfacial compatibility between the layers and the integrity of the 3D printing process. From the above discussions, it could be observed that the infilling of the honeycomb pattern with another infill contributed significantly towards the enhancement of the mechanical properties of the 3D-printed PLA/bamboo biocomposites.

#### 3.2.2. Hardness Test

A Shore-D durometer was used to conduct hardness tests for the 3D printed biocomposite specimen. Table 3 reveals the values of hardness in Shore D, and Figure 6 the honeycomb infill grid pattern. The hardness of the grid infill honeycomb revealed the hardness value of 74 Shore D. The test reveals that the grid pattern’s squares and rectangles are formed by straight, intersecting lines that distribute the applied loads evenly. The overall strength is enhanced, because this uniformity prevents any one area from being overstressed. The regular and continuous paths of the grid pattern improve the bonding between the layers. Composites with bamboo fibers have higher tensile and compressive strengths, so it is crucial that the layers adhere well to one another. In order to minimize gaps and weak points, grid infill is useful for designs that are irregular or discontinuous, like Gyroid or Concentric. Stress can be transferred directly along the grid’s intersecting lines. More complex infill patterns, such as Cubic Subdivision or Tri-hexagon, can have curved or angled lines, which hinder the structure’s force transfer. A linear grid, on the other hand, is ideal. The X and Y axes, which are perpendicular to each other, are inherently reinforced by grid patterns. Bidirectional reinforcement is superior to infill patterns such as Zig Zag or Triangle because it increases the specimen’s resistance to deformation under multidirectional loads.

### 3.3. Effect of Infill Pattern on Tribological Properties of PLA/Bamboo Composites

Wear and frictional force were measured on a Pin-on-Disc tester in accordance with the ASTM standards, and the SWR was calculated based on the above values. In addition, Table 5 reveals the values of wear behavior. It infers that the composite specimens made of 3D-printed PLA and bamboo feature an Octet infill honeycomb pattern that is exceptionally robust due to its well-designed 3D lattice geometry, which integrates diagonal and vertical struts to achieve isotropic strength and uniform load distribution. With its minimal material usage and high strength-to-weight efficiency, this design offers strong resistance to buckling and multi-axial stresses. Energy absorption, interlayer bonding, and dynamic load resistance are all enhanced by the PLA matrix by utilizing the tensile and bending properties of bamboo fibers. Elements with triangle and diagonal connections are used for this purpose. When it comes to mechanical performance and material loss, the Octet honeycomb pattern is the best.

## 4. Conclusions

PLA/bamboo biocomposites were 3D-printed with various infill patterns, and the experimental work reveals the following conclusions and leads to various implications.

→FTIR results exhibited the presence of cellulose and lignin, portraying the dispersion of bamboo in PLA. X-ray diffraction analysis revealed that the crystallite size of the composites was 23 nm. The presence of condensed particles in the composites aided in better load-carrying capacity by offering better stress transfer, resulting in better mechanical properties of the biocomposites.→Impact test results showed that the cross infill pattern, with its crisscross stress distribution, had the greatest impact strength (6.51 J/cm^2^) due to less stress, and localized failures occur when lines intersect regularly due to the improved interlayer bonding. When compared to irregular geometries such as Gyroid and Tri-hexagon, cross infill patterns demonstrate superior impact resistance.→Morphological analysis of the impact failure specimen portrayed that the interfacial bonding between the layers and the materials was commendable, which contributed to the least failure initiation during impact load. Infilling of the honeycomb structure with another infill pattern contributed towards the load sharing and effective stress transfer during the loading.→Hardness test results showed that the pattern with the grid infill had the greatest Shore D hardness, at 74. It follows that the grid’s consistent square shape enhances load distribution and adhesion between layers, and the resistance to deformation was enhanced by using bidirectional reinforcement in grid patterns. Reduced structural efficiency is observed in patterns with curved or angled lines, such as Gyroid or Cubic Subdivision.→The Octet infill pattern exhibited remarkable wear resistance and minimal material loss during the wear test. The strong three-dimensional lattice structure incorporates both vertical and diagonal struts to achieve isotropic strength. Reduced damage from dynamic loads and improved energy absorption are both benefits of a high strength-to-weight efficiency. The structural robustness and wear resistance of PLA/bamboo composites are enhanced using diagonal and triangular connections.

The developed PLA/bamboo biocomposites find their applications in load-carrying structures in automobile and aerospace components. Further works can be focused on developing a specific application component using these materials, and testing of the real-time application can be carried out. In addition, this work provides a path to researchers in this domain to gain insights to choose the infill pattern to fill the honeycomb structure based on the application; further work can be extended by increasing the number of honeycombs in the specimen.

## Figures and Tables

**Figure 1 polymers-17-00448-f001:**
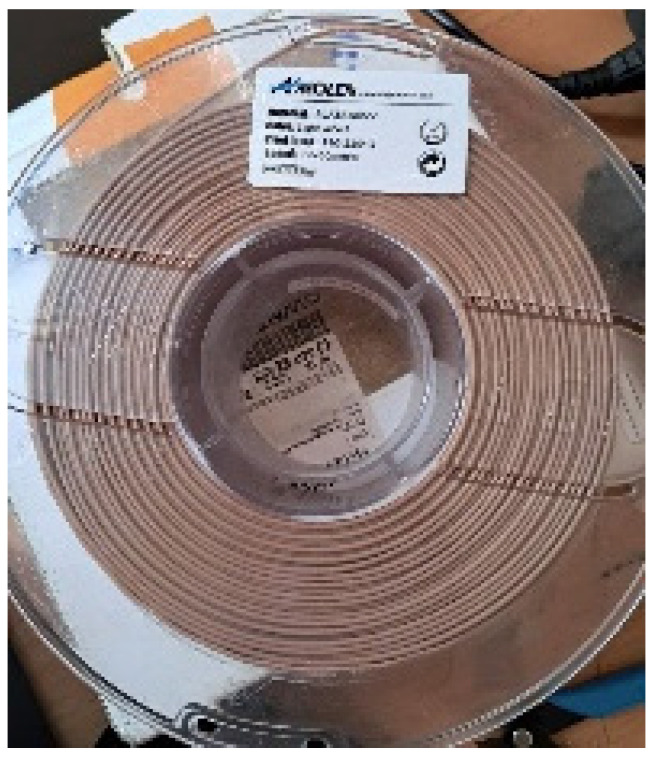
PLA/bamboo filament.

**Figure 2 polymers-17-00448-f002:**
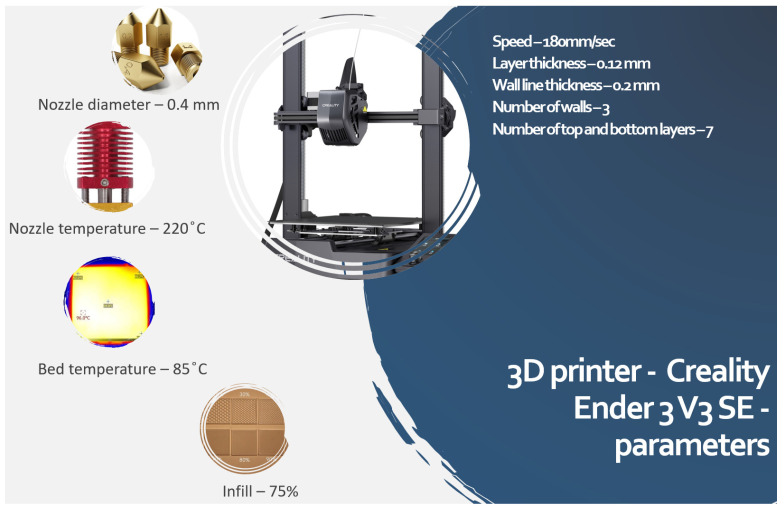
3D printing parameters.

**Figure 3 polymers-17-00448-f003:**
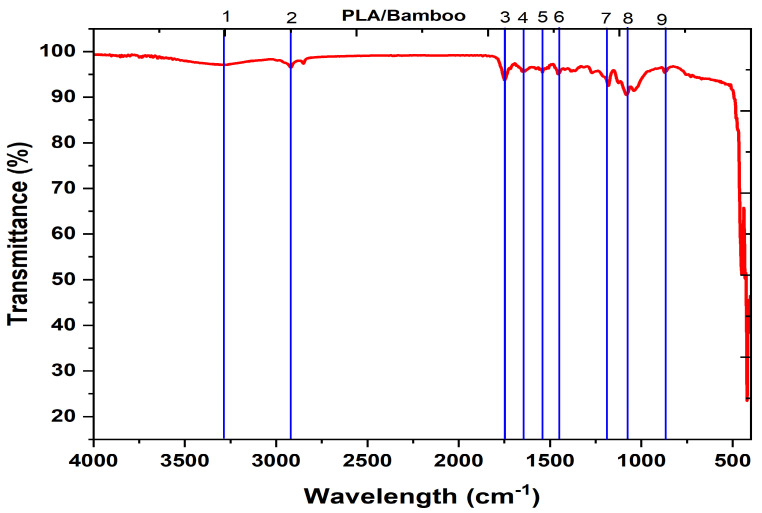
FTIR peaks for PLA/bamboo composite filament.

**Figure 4 polymers-17-00448-f004:**
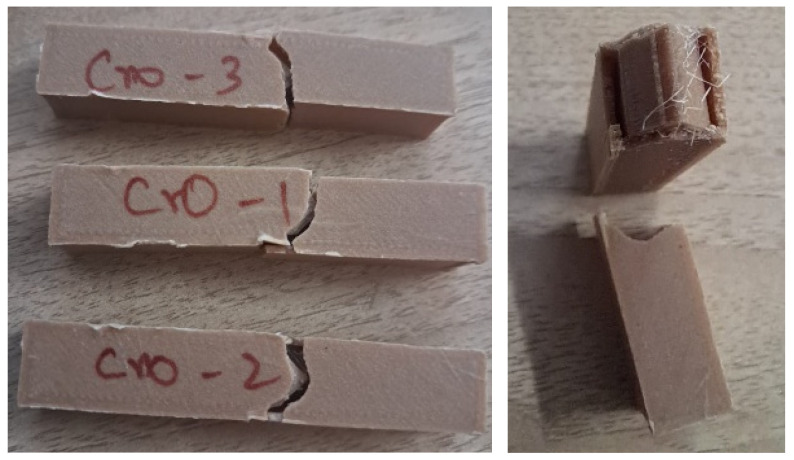
Impact specimen of cross infill honeycomb after test.

**Figure 5 polymers-17-00448-f005:**
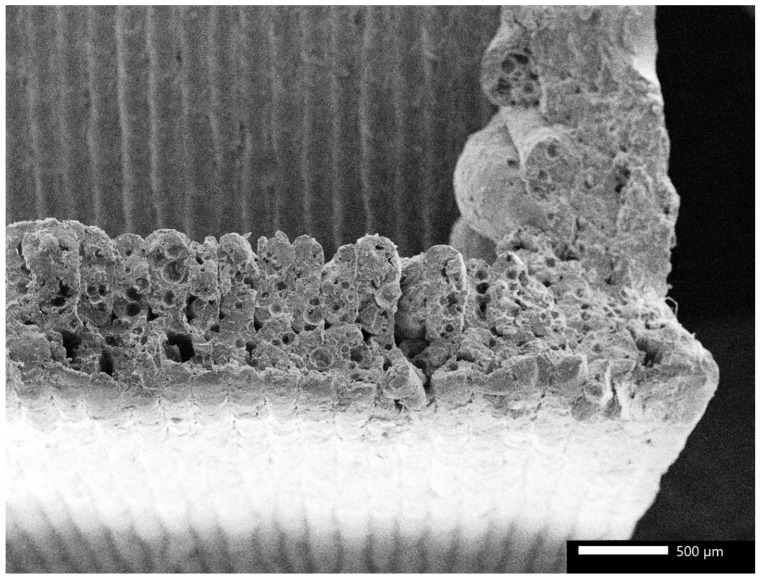
Morphology of PLA/bamboo specimen after impact failure.

**Figure 6 polymers-17-00448-f006:**
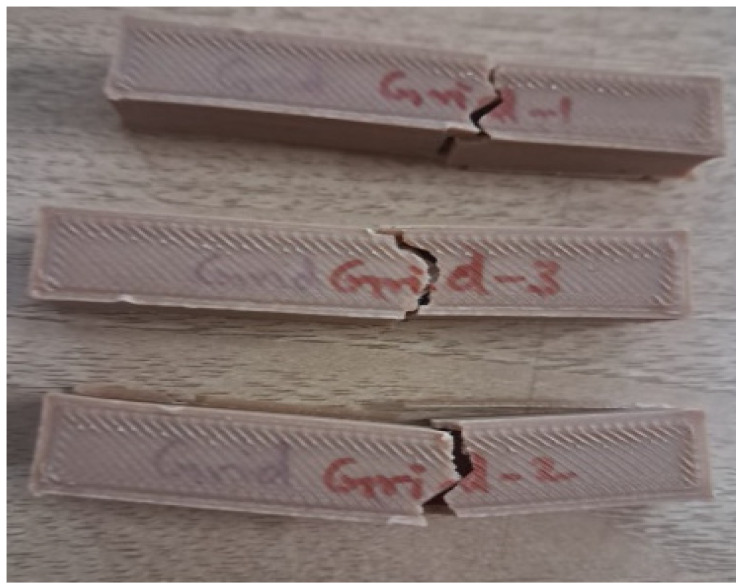
Hardness specimen of grid infill honeycomb after test.

**Table 1 polymers-17-00448-t001:** Properties of PLA.

S. No.	Description and Units	Value Range
1	Density (g/cm^3^)	1.25 g/cc
2	Tensile Strength (MPa)	51–70
3	Impact Energy absorption (J/cm^2^)	2.1–5.1
4	Hardness	60–70

**Table 2 polymers-17-00448-t002:** Mechanical and tribological test specimen details.

S. No.	Testing Method	ASTM Standard [23,24,25]	Specimen Dimension (mm^3^)	Remarks
1	Impact test	ASTM D256	64 × 12.7 × 3.1	Load on the hammer = 5.5 J, Striking angle = 140°
2	Hardness test	ASTM D2240	10 × 10 × 10	Dead weight = 50 N
3	Wear test	ASTM G99	30 × 10 × 10	Load—10 N, Speed—190, 380 and 570 rpm, Time—1200 s

**Table 3 polymers-17-00448-t003:** FTIR Results.

Peak Numbers	Wavelength Range (cm^−1^)	Description
1, 2	4000–3000	O-H bond and N-H bonds
1	3400	Hydroxyl (O-H) vibrations indicating cellulose and lignin
2	2800	C-H stretching from aliphatic chains
3, 4	1750–1735	Ester carbonyl (C=O) stretch
5–9	<1500	Bending and stretching of C-H and C-O bonds

**Table 4 polymers-17-00448-t004:** Mechanical Test Results.

S. No	Infill Pattern	Shore D Hardness	Impact Strength in J/cm^2^
1	Quadra Cubic	59.17	5.95
2	Grid	74.00	1.08
3	Tri-hexagon	64.33	5.63
4	Cubic Subdivision	72.83	1.91
5	Concentric	67.67	6.17
6	Octet	67.17	5.39
7	Cross	68.83	6.51
8	Triangle	45.45	0.92
9	Zig Zag	72.67	6.06
10	Cross 3D	67.50	1.16
11	Gyroid	67.67	0.83
12	Cubic	63.83	4.46

**Table 5 polymers-17-00448-t005:** Comparison of wear behavior of specimens (load = 10 N; time = 1200 s).

S. No	Infill Pattern	Speed (rpm)	Frictional Force in N	Height Loss in µ	SWR (mm^3^/Nm)	Average SWR (mm^3^/Nm)
1	Octet	190	3.2	223	0.012469	0.007584
		380	7.3	580	0.006162	
		570	6.4	488	0.00412	
2	Concentric	190	8.6	407	0.012392	0.007543
		380	7.7	710	0.006134	
		570	6.3	606	0.004104	
3	Cubic Subdivision	190	6.8	348	0.012417	0.007561
		380	5.3	512	0.006176	
		570	5.8	712	0.004089	
4	Quadra Cubic	190	6.3	410	0.012391	0.007539
		380	7.4	556	0.006167	
		570	8.5	927	0.004059	
5	Grid	190	6.0	567	0.012325	0.007519
		380	7.9	743	0.006127	
		570	4.8	596	0.004105	
6	Triangle	190	6.2	512	0.012348	0.00752
		380	5.6	572	0.006163	
		570	8.0	1000	0.004049	
7	Cross 3D	190	5.4	410	0.012391	0.007565
		380	4.6	507	0.006177	
		570	3.5	437	0.004127	
8	Tri-hexagon	190	6.8	408	0.012392	0.00755
		380	7.0	480	0.006182	
		570	9.0	800	0.004077	
9	Cross	190	6.5	320	0.012429	0.007555
		380	4.9	570	0.006164	
		570	4.5	826	0.004073	
10	Gyroid	190	8.6	773	0.012239	0.007514
		380	4.7	487	0.006181	
		570	5.4	484	0.004121	
11	Cubic	190	4.6	576	0.012322	0.007511
		380	6.6	550	0.006168	
		570	11.2	1035	0.004044	
12	Zig Zag	190	7.3	292	0.012441	0.007556
		380	6.4	550	0.006168	
		570	12.1	920	0.00406	

## Data Availability

The original contributions presented in this study are included in the article. Further inquiries can be directed to the corresponding authors.

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
