# Peer review of "Effects of Infill Patterns on the Mechanical and Tribological Behaviour of 3D-Printed Polylactic Acid/Bamboo Biocomposites for Structural Applications"

_polymers, 2025, doi:10.3390/polym17040448_

Round 1

Reviewer 1 Report

Comments and Suggestions for Authors

The article examines composite materials produced using polylactic acid (PLA) and bamboo fibers (BF) in a honeycomb structure with various infill patterns. In the study, a composite material prepared with a ratio of 80% PLA and 20% bamboo fiber was produced using a 3D printing method. Twelve different infill patterns were applied to the honeycomb structure, and the mechanical properties of these structures were evaluated.

  1. The introduction includes a literature review, but it does not clearly state the gaps in existing studies or how this study fills those gaps. The strengths of the article should be expressed more explicitly.

  2. It has not been clarified how this study differs from other research. More precise statements explaining this distinction should be provided.

  3. The main objective of the research is unclear. The specific questions being addressed (e.g., which infill pattern improves which mechanical property) should be articulated more clearly.

  4. The dimensions of the tested specimens (length, width, thickness) should be explicitly stated.

  5. The mechanical or chemical properties of PLA and bamboo fibers should be provided.

  6. The exact ASTM standard numbers have not been mentioned. Complete references to these standards should be included.

  7. The environmental conditions under which the study was conducted (e.g., temperature, humidity) have not been specified. Such conditions could significantly affect the results of mechanical tests.

  8. The sample preparation process (e.g., whether the filament was dried or underwent any special treatment before printing) is missing.

  9. There is no control group, such as pure PLA, to compare the performance of composite samples made with PLA and bamboo fibers. This would have provided a clearer understanding of the effects of different infill patterns.

Author Response

REVIEWER 1

Thank you for spending your to review the article and improve the quality of it and authors sincerely appreciate your effort.

Comment – 1:-

The introduction includes a literature review, but it does not clearly state the gaps in existing studies or how this study fills those gaps. The strengths of the article should be expressed more explicitly.

Response – 1:-

As per the suggestion of the reviewer, the gaps identified from the literature has been included in the revised manuscript. The strength of the article is explicitly included in the introduction section.

Comment – 2:-

It has not been clarified how this study differs from other research. More precise statements explaining this distinction should be provided.

Response – 2:-

The suggestion of the reviewer is well-taken. The novelty statement of the article has been included in the introduction section of the revised manuscript.

Comment – 3:-

The main objective of the research is unclear. The specific questions being addressed (e.g., which infill pattern improves which mechanical property) should be articulated more clearly.

Response – 3:-

As per the suggestion of the reviewer, the objective of the work has been included in the revised manuscript.

Comment – 4:-

The dimensions of the tested specimens (length, width, thickness) should be explicitly stated.

Response – 4:-

The suggestion of the reviewer is well-taken. Table 2 of the revised manuscript contains the test details, testing conditions and specimen details.

Comment – 5:-

The mechanical or chemical properties of PLA and bamboo fibers should be provided.

Response – 5:-

As per the suggestion of the reviewer, Table 1 of the revised manuscript enlists the general properties of PLA.

Comment – 6:-

The exact ASTM standard numbers have not been mentioned. Complete references to these standards should be included.

Response – 6:-

The suggestion of the reviewer is well-taken. Table 2 of the revised manuscript contains the test details, testing conditions and specimen details.

Comment – 7:-

The environmental conditions under which the study was conducted (e.g., temperature, humidity) have not been specified. Such conditions could significantly affect the results of mechanical tests.

Response – 7:-

As per the suggestion of the reviewer, the environmental conditions for 3D printing of the sample and testing was included under section 2.2 and section 2.3.1 respectively.

Comment – 8:-

The sample preparation process (e.g., whether the filament was dried or underwent any special treatment before printing) is missing.

Response – 8:-

The suggestion of the reviewer is well-taken. However, the PLA/Bamboo filaments were not made by the authors but was purchased as such from M/S. AMOLEN, an USA based company.

Comment – 9:-

There is no control group, such as pure PLA, to compare the performance of composite samples made with PLA and bamboo fibers. This would have provided a clearer understanding of the effects of different infill patterns.

Response – 9:-

As per the suggestion of the reviewer, the general properties of the pure PLA was included as table 1 in the revised manuscript. It could be generally found that the mechanical properties of the 3D printed PLA/Bamboo composites were more or less on par with the control samples.

Reviewer 2 Report

Comments and Suggestions for Authors

A review for paper:

Poly Lactic Acid Bamboo Composite is Analysed for Honeycomb Structure with
Various Infill Pattern

by D. Balaji, B. Arulmurugan and V. Bhuvanewari

Paper ID: polymers-3440312

The paper entitled „Poly Lactic Acid Bamboo Composite is Analysed for Honeycomb Structure with Various Infill Pattern” describes preparation of 3D printed parts made of PLA/bamboo fibers composite and characterization of these materials according their mechanical stability, structure and crystallinity (FTIR, XRD).

The paper is poorly written, as it lacks sufficient details regarding testing, figures and the English corrections are necessary before publication. The comments are correct however their description should be revised, as now there are not clear and difficult to understand. The paper has proper structure and appropriate references, however some parts of the paper should be reformulated as for example Abstract and Conclusions part.

The paper in the present form cannot be published in the Polymers and major revision is suggested.

Here are my general comments regarding paper:

1.      Abstract – should be concise and report on the research design (methods) and the most important findings. Please, consider to reformulate the abstract as in the present form is not so clear, for example there is no information about performed testing for the materials.

2.      Fig.2 it should be presentation of process parameters nor the printer brochure from the manufacturer. Please consider to move this information back to the text. Better to add information regarding the patterns that were produced (visual information), as well as table with samples designation.

3.      Materials and methods – there should be information regarding testing methods, for example IR procedure, parameters for XRD analysis and mechanical testing procedures, standards, sample size. The minimal number of ”trials” for mechanical testing is usually five to ensure the reliability of test results, please explanation why only 3 were mentioned here.

4.      Table 2, page 5 – are there 2 methods as stated in Table heading? Please correct.

5.      Table 3, page 6 – please put all values which are the same for samples (e.g. Load, time) in the table heading to limit table size and for clarity. Moreover, as there is no description of characterization for the samples it is not clear how the authors get these values of frictional force and height loss. As mentioned before, the minimal amount of samples for reliable test results is 5. Moreover, usually the wear test is followed by analysis with a surface profilometer and a microscope to observe the wear surfaces and possibles changes. In this sense the Figure 5 is useless as it is not informative.

6.      Page 7, line 153 “to the oxyl (O-H)” – this is hydroxyl group and respective vibration. Please correct. Moreover, fig.6 represents assigned IR peaks without detailed description, please consider to add table with description of signals and their assignment to different characteristic vibrations. Please improve the image quality, there is no need to leave information from the software about evaluation technique (peak find). Not every peak is visible, i.e. peaks 11-13, please add some zoom in of this area or just eliminate these signals. The band related to carbonyl group is of very low intensity which is weird because it is usually the most intensive band.

7.      Page 8, line 163 “(23.08 nm)” – what this value means? The size of bamboo fibers inside the PLA matrix or sth else? Please be more clear with such information. The authors stated that PLA crystallinity is higher however there’s no proper comparison or reference to give such statement. Higher than what? Pure PLA? Please correct. The sharp peaks in XRD means fine crystals formation, the crystallinity is calculated with particular equation using the peaks “size”, and this equation and the procedure of crystallinity calculation should be presented. Please, consider to add XRD diffractogram, as it will help to assess the results. Please correct the Table 4 – the similar numbers (with similar amount of significant numbers).

8.      There is no justification for conducting XRD analysis for PLA filament. Why was it decided to conduct this test when it does not contribute anything to the materials produced during the research?

9.      Conclusions – please reformulate this part, conclusions come from the research, there are no requirements. Conclusions should be sth like summary of the observed effects and the authors explanation for this, not the repetition of “results and discussion part”. For example FTIR results – there is no proof in the text that the chemical compatibility between PLA and bamboo exists. This should be in the discussion part, here in conclusion should be only general conclusion.

10.   Page 9, lines 202-207 – “Imaging by X-Ray Diffraction”, XRD method is not related to morphology, so the “imaging” is not proper. There’s no XRD diffractogram involved so the reader cannot see the peaks. There’s no reference material, for example XRD data for pure PLA and bamboo fibers, which will be helpful to observe change in the crystallinity. It should be noted that PLA should be processed in the same way as PLA filament.

Comments on the Quality of English Language

Please perfomr carefull English checking. Now some senetnces are short and unclear. Chcekc the typos e.g. page 5 line 133 "in filled honeycomb".

Author Response

REVIEWER 2

Thank you for spending your to review the article and improve the quality the article, authors sincerely appreciate your effort.

Comment – 1:-

Abstract – should be concise and report on the research design (methods) and the most important findings. Please, consider to reformulate the abstract as in the present form is not so clear, for example there is no information about performed testing for the materials.

Response – 1:-

As per the suggestion of the reviewer, the abstract has been reframed by including the research design and testing methods of the composites.

Comment – 2:-

Fig.2 it should be presentation of process parameters nor the printer brochure from the manufacturer. Please consider to move this information back to the text. Better to add information regarding the patterns that were produced (visual information), as well as table with samples designation.

Response – 2:-

The suggestion of the reviewer is well taken. Figure 2 is a self-drawn figure which portrays the 3D printing process parameters and the 3D printer used for fabricating the test specimen. All the test specimens were of same composition since the same PLA/Bamboo filament was used and so sample designations for the tests were kept in the names of infill patterns itself.

Comment – 3:-

Materials and methods – there should be information regarding testing methods, for example IR procedure, parameters for XRD analysis and mechanical testing procedures, standards, sample size. The minimal number of “trials” for mechanical testing is usually five to ensure the reliability of test results, please explanation why only 3 were mentioned here.

Response – 3:-

As per the suggestion of the reviewer, section 2.3 gives the details of the mechanical and tribological tests and the details of characterization methods.

Comment – 4:-

Table 2, page 5 – are there 2 methods as stated in Table heading? Please correct.

Response – 4:-

As per the suggestion of the reviewer, the table heading has been corrected in the revised manuscript.

Comment – 5:-

Table 3, page 6 – please put all values which are the same for samples (e.g. Load, time) in the table heading to limit table size and for clarity. Moreover, as there is no description of characterization for the samples it is not clear how the authors get these values of frictional force and height loss. As mentioned before, the minimal amount of samples for reliable test results is 5. Moreover, usually the wear test is followed by analysis with a surface profilometer and a microscope to observe the wear surfaces and possible changes. In this sense the Figure 5 is useless as it is not informative.

Response – 5:-

The suggestion of the reviewer is well-taken. Section 2.3.1 enunciates the testing method and the method of obtaining values from the tribometer. A total of 3 trials were taken for averaging wear and frictional force considering the availability of filament and the machine. Authors kindly request the reviewers to consider this for further processing of the manuscript. Figure 5 has been removed from the revised manuscript.

Comment – 6:-

Page 7, line 153 “to the oxyl (O-H)” – this is hydroxyl group and respective vibration. Please correct. Moreover, fig.6 represents assigned IR peaks without detailed description, please consider to add table with description of signals and their assignment to different characteristic vibrations. Please improve the image quality, there is no need to leave information from the software about evaluation technique (peak find). Not every peak is visible, i.e. peaks 11-13, please add some zoom in of this area or just eliminate these signals. The band related to carbonyl group is of very low intensity which is weird because it is usually the most intensive band.

Response – 6:-

As per the suggestion of the reviewer, the image quality has been improved and a table with the description of peaks has been included in the revised manuscript.

Comment – 7:-

Page 8, line 163 “(23.08 nm)” – what this value means? The size of bamboo fibers inside the PLA matrix or sth else? Please be clearer with such information. The authors stated that PLA crystallinity is higher however there’s no proper comparison or reference to give such statement. Higher than what? Pure PLA? Please correct. The sharp peaks in XRD means fine crystals formation, the crystallinity is calculated with particular equation using the peaks “size”, and this equation and the procedure of crystallinity calculation should be presented. Please, consider to add XRD diffractogram, as it will help to assess the results. Please correct the Table 4 – the similar numbers (with similar amount of significant numbers).

Response – 7:-

As per the suggestion of the reviewer, the inference drawn from the XRD data has been rewritten for clarity with suitable references.

Comment – 8:-

There is no justification for conducting XRD analysis for PLA filament. Why was it decided to conduct this test when it does not contribute anything to the materials produced during the research?

Response – 8:-

XRD analysis was carried out to calculate the crystallite size of the composite. Crystallite size of the composite was related with the mechanical properties of the composite. The same has been included in the revised manuscript under section 3.1.2.

Comment – 9:-

Conclusions – please reformulate this part, conclusions come from the research, and there are no requirements. Conclusions should be sth like summary of the observed effects and the author’s explanation for this, not the repetition of “results and discussion part”. For example FTIR results – there is no proof in the text that the chemical compatibility between PLA and bamboo exists. This should be in the discussion part, here in conclusion should be only general conclusion.

Response – 9:-

As per the suggestion of the reviewer, the conclusion section has been reframed in the revised manuscript.

Comment – 10:-

Page 9, lines 202-207 – “Imaging by X-Ray Diffraction”, XRD method is not related to morphology, so the “imaging” is not proper. There’s no XRD diffractogram involved so the reader cannot see the peaks. There’s no reference material, for example XRD data for pure PLA and bamboo fibers, which will be helpful to observe change in the crystallinity. It should be noted that PLA should be processed in the same way as PLA filament.

Response – 10:-

As per the suggestion of the reviewer, the XRD data in the conclusion section has been rewritten thoroughly for clarity.

Round 2

Reviewer 2 Report

Comments and Suggestions for Authors

Dear Authors,

thanks for the corrections.

I think that it is possible to improve Fig3 quality y redrawing it in separate program rather than loading the original scan from the software.

Please correct in table 3: hydroxyl group not oxyl group.

Table 4 was not corrected, as I suggested.

I accept the paper after these small corrections.

Author Response

RESPONSE TO REVIEWERS’ COMMENTS

Thank you so much for the editor to provide the author to amend the article and thank you so much for the reviewer to spend time on our article to improvise it.

Comment 1:

I think it is possible to improve Fig 3 quality by redrawing it in a separate program rather than loading the original scan from the software

Response 1:

As per the suggestion of the reviewer, the FTIR plot has been redrawn using a dedicated plotting software to improve its readability.

Comment 2:

Please correct in table 3: hydroxyl group not oxyl group

Response 2:

As per the suggestion of the reviewer, the correction has been amended in the revision.

Comment 3:

Table 4 was not corrected as I suggested

Response 3:

As per the suggestion of the reviewer, table 4 was corrected by including the common data in the table caption to reduce the table size.
